# HPV Vaccine: An Effective but Underutilized Prevention Tool

**DOI:** 10.3390/ijerph22121844

**Published:** 2025-12-10

**Authors:** Vittorio Grieco, Debora Fiorito, Gabriele Giorgianni, Eleonora Irato, Alessia Anna Di Prima, Chiara Chillari, Antonella Ippolito, Liliana Mereu, Rosalia Ragusa

**Affiliations:** 1Department of Medical, Surgical Sciences and Advanced Technologies, University of Catania, 95123 Catania, Italy; commissionehta@policlinico.unict.it (V.G.); chiara.chillari@asp.messina.it (C.C.); 2Gynecologic and Obstetrics Unit, Azienda Ospedaliero Universitaria Policlinico “G. Rodolico-San Marco”, 95123 Catania, Italy; ricerca@policlinico.unict.it (D.F.); liliana.mereu@unict.it (L.M.); 3Department of Prevention, UOS Coordinamento Operativo delle Attività Vaccinali Metropolitane e Provinciali–ASP 3 Catania, 95027 Catania, Italy; gabriele.giorgianni@aspct.it; 4Integrated Cancer Registry CT-ME-EN, Azienda Ospedaliero Universitaria Policlinico “G. Rodolico-San Marco”, 95123 Catania, Italy; e.irato@policlinico.unict.it (E.I.); a.diprima@policlinico.unict.it (A.A.D.P.); 5School of Specialization in Health Statistics and Biometrics, University of Pavia, 27100 Pavia, Italy; antonellaippolito81@gmail.com; 6HTA Committee, Azienda Ospedaliero Universitaria Policlinico “G. Rodolico-San Marco”, 95123 Catania, Italy

**Keywords:** HPV, human papillomavirus, HPV infection, HPV vaccine, cervical cancer, adjuvant vaccination, cervical lesions

## Abstract

HPV vaccines are among the most effective vaccines available, offering safe administration and high cost-effectiveness. The composition of the vaccines has been changed, involving enrichment with pathogenic strains and extending the possibility of prevention to other HPV-related cancers and diseases. The efficacy of the vaccine extends beyond primary prevention due to documented reductions in the relapse of cervical lesions. The objective of the present study was to evaluate whether there were differences in the frequency of high-grade squamous intraepithelial lesions (H-SILs) and cervical carcinoma between vaccinated and unvaccinated women. This retrospective study included all incident cases of cervical cancer or H-SIL diagnosed between 1 January 2003 and 31 December 2020, in Catania, Italy. We analyzed 2722 cases: 751 cervical cancers and 1901 H-SILs. A total of 88% of patients had never been vaccinated. Among women with H-SILs, 85% had never received a vaccination against HPV. Among the few women with H-SILs who were vaccinated, 3.5% of them received at least one dose of the HPV vaccine before diagnosis, while 96.5% started the HPV vaccine cycle after diagnosis of cervical lesions. The mean age at diagnosis for our population was 43.4 years (52 years for cervical cancer; 39.8 years for H-SIL). Statistically significant differences were found between vaccinated and unvaccinated women. The HPV vaccine is an underutilized intervention that has the potential to eradicate cervical cancer and reduce the occurrence of other HPV-related cancers. Implementing new communication strategies can increase the number of vaccinated subjects.

## 1. Introduction

The eradication of cervical cancer is an important goal in the fight against cancer, as it would be the first type of cancer to be potentially eliminated as a public health problem. This cancer shows bimodal age distribution with the first peak at 30–34 years of age, due to first Human Papillomavirus (HPV) infection after sexual initiation and a second peak at 65–69 years of age due to viral persistence or reactivation of latent HPV [1]. According to the WHO’s Cervical Cancer Elimination Initiative, three key targets must be reached by 2030 to eliminate cervical cancer as a public health problem: 90% of girls fully vaccinated against HPV by the age of 15; 70% of women screened using a high-performance test by the age of 35 and again by the age of 45; and 90% of women with adequately treated cervical lesions and invasive cancers [2].

In 2021, the European Commission furthered the initiative through Europe’s Beating Cancer Plan, raising cervical screening coverage to 90% for member countries of the European Union [3].

HPV vaccination coverage varies significantly across Europe, ranging from less than 5% to over 90%. According to the official report by the WHO in 2023, the mean coverage of the HPV vaccination program, i.e., completion of last dose, among women in the WHO European region was 36%. In Turkmenistan, Uzbekistan, Norway, Cyprus, Portugal, and Sweden, the coverage exceeded 85% of the population, while in Bosnia, Bulgaria, Serbia, and Poland, the official coverage was below 5% [4].

In line with other EU countries, Italy implemented national vaccination programs and introduced HPV vaccinations free of charge for girls aged 11–12 years in 2008; an out-of-pocket payment would be required for older ages. The Italian Vaccination Prevention Plan provides free vaccination with a nonavalent vaccine during the 12th year of age for both females and males.

The estimated number of new cervical cancer diagnoses in Italy in 2024 was approximately 2382 [5]. The catch-up program provides all doses of the vaccination cycle, free of charge, for women up to 26 years of age and for men up to and including 18 years of age, if they have not previously been vaccinated or have not completed the vaccination cycle. Furthermore, the vaccine is free for women suffering from cervical lesions/HpvDNA+, women with risk conditions upon prescription, and men whose partners are HPV-positive women or at risk, always upon prescription. The average Italian vaccination coverage varies over time. In the first cohorts included (2000), vaccination was completed in 68% of girls; for the 2012 cohort of girls, an average Italian coverage of 51% is achieved with large differences observed between the various regions (range 78–23%) [6].

Studies have confirmed over time the efficacy of various vaccines authorized on the market, as well as their safety and cost-effectiveness [7,8,9]. In recent years, vaccine compositions have changed, following enrichment with pathogenic strains, and the possibility of prevention has extended to other HPV-related cancers and diseases [10,11,12,13]. Increased coverage of HPV vaccines can significantly reduce the risk of other urogenital and head–neck cancers and benign lesions such as genital warts [14,15,16]. Vaccinating patients with advanced cervical lesions may be an effective way to reduce recurrent H-SILs [17] and prevent cervical cancer, even though no therapeutic HPV vaccine has yet been approved for H-SILs and the use of lentiviral vaccines, although promising, is still in the preclinical phase [18].

Assessing HPV vaccination rates among women with high-grade cervical lesions or cancer allows identification of gaps in prevention, evaluates patterns of vaccine uptake, and informs strategies to enhance coverage and advance elimination initiatives.

In our study, we aimed to determine how many women with cervical cancer or high-grade dysplasia were previously vaccinated and how many women had received the vaccine after these diagnoses.

## 2. Materials and Methods

This retrospective observational study was conducted between 15 October 2024, and 31 January 2025, by the Health Technology Assessment Committee of Policlinico “G. Rodolico- San Marco”, a reference teaching and research hospital in Catania, Italy.

We collected information on women residing in the province of Catania, with a population of approximately one million inhabitants, who had a confirmed cytological or histological diagnosis of H-SIL or cervical cancer between 1 January 2003, and 31 December 2020. We examined the number of women who had received the HPV vaccination. Patients who passed away before the introduction of HPV vaccination in Sicily (28 April 2008) were excluded. Information on cervical cancer and high-grade cervical lesions was obtained from the Integrated Cancer Registry of Catania-Messina-Enna.

The data on cervical cancer and other cervical lesions were cross-checked between cases registered in the local vaccination registry of the Province of Catania and in the National Register of Causes of Death (RENCAM).

Patients were included if they met the following criteria: cytological or histological diagnosis of HPV-related cervical cancer or high-grade cervical lesions; residency in the province of Catania during the study period; and presence in the local vaccination registry. Cytological evaluations followed the Bethesda System for Reporting Cervical Cytology [19], while biopsy samples were classified based on the WHO Classification of Tumors, as outlined by the International Agency for Research on Cancer (IARC). Morphology was coded according to the International Classification of Disease for Oncology, third edition (ICDO-3) [20].

For each patient, data were collected on diagnosis date, age at diagnosis, type and classification of the cancer or lesion, and tumor stage at the time of diagnosis. HPV genotyping data (HPV genotyping) were not analyzed in the present study.

Vaccination status was determined using the Provincial Vaccination Registry of Catania, where all vaccinations performed after 2008 were registered. A case verification agreement was established with the Prevention Department of the local health authority (Provincial Agency for Health of Catania—ASP 3). Vaccination has always been performed after a prescription and cannot be performed privately. All HPV vaccines have been registered with the ASP 3 since the start of the vaccine offering coincides with the computerization of data. Every doctor is obliged to upload the vaccination to the dedicated portal. Data extracted included vaccine type, number of doses administered, and administration dates. In Italy, two doses are recommended for individuals up to 15 years old, and three doses for those above 15 years [21].

We categorized our patients as follows: fully vaccinated, if they received recommended doses of the vaccine; partially vaccinated, if they received fewer than the recommended doses; and not vaccinated if they received no doses. We compared the dates of diagnosis of carcinoma and H-SIL, obtained from pathological anatomy reports, with the dates of vaccine administration, provided by the ASP 3. The patients were sub-categorized into two groups: those who had a vaccination before the diagnosis and those who had a vaccination after the diagnosis.

The tetravalent (4-serotype, 4vHPV) HPV vaccine was introduced in Catania in April 2008 and was later replaced by the nonavalent (9-serotype, 9vHPV) vaccine in September 2017. Patients’ living status was verified using the RENCAM, updated as of 31 December 2023. Diseases occurring 18 months after the first vaccine administration were considered to be vaccine failures.

### 2.1. Statistical Analysis

Data from the three sources were linked using a unique identification code assigned to each patient, ensuring anonymity and compliance with data protection regulations.

Statistical analyses were performed using R version 4.5.2.

To describe vaccination status distribution, the following parameters were mean age at diagnosis, number of vaccine doses received at the time of diagnosis, number of vaccines administered post-diagnosis, mean and median time between diagnosis and vaccination, vaccination coverage among eligible patients, and the fraction of patients vaccinated after diagnosis. Vaccination status and age were compared between subgroups based on tumor, distinguishing between malignant tumors and H-SILs.

To determine whether there were statistically significant differences in the frequency of H-SIL and cervical carcinoma between vaccinated and unvaccinated women, we used Pearson’s chi-square test. When expected frequencies were <5 or a cell contained a zero value, statistical significance was verified using the two-tailed Fisher’s exact test. In the presence of zero events, the Haldane–Anscombe correction was applied to obtain a stable estimate of the relative risk (RR). A *p*-value < 0.05 was considered statistically significant.

### 2.2. Reporting

We reported the study characteristics according to the STROBE checklist for cohort studies [22] (Appendix A).

## 3. Results

We analyzed 2722 cases of women living in the province of Catania, Sicily, Italy, diagnosed with cervical tumor or H-SIL between 2003 and 2020 (Figure 1).

As the first HPV vaccine was available in Catania in April 2008, 70 women who had died before the vaccine became available were excluded from the study. As we can see in Figure 1, the number of incident cases of cervical cancer, coded by the Integrated Cancer Registry of the Provinces of Catania, Messina, and Enna, although with slight fluctuations, does not show a significant reduction in cases over the years. Conversely, since 2013, when cervical cancer screening switched from cytology to HPV testing using the validated PCR-based technique, in combination with the PAP test, HPV positivity and detection rate of H-SIL have shown a significant increase.

HPV vaccination was introduced in our region in 2008. The Italian vaccination schedule provides for the administration of two doses to girls under 15 and three doses to those over 15.

The number of women fully and partially vaccinated among the resident population of Catania and the number of fully and partially vaccinated women with cervical abnormalities are reported in Figure 2.

The number of vaccinated women in the resident population and the number of vaccinated women with cervical abnormalities remained very low until 2017. A small increase is noted in women with cervical abnormalities coinciding with the introduction of HPV testing for human papillomavirus genotyping and cervical precancer screening. Later, evidence shows that the infection is very common in the population [23] and gave impetus to vaccination. In our province, a clear increase in the vaccinated population has been recorded in recent years, even if the methods and number of doses are not always correct. HPV vaccine course completion and dose adherence were faster and higher among women with cervical abnormalities than in the general healthy population.

Our study included 2652 women: 751 (28.3%; 95% CI: 26.6–30.0) with cervical cancer and 1901 (71.7%; 95% CI: 70.0–73.4) with H-SILs. Only 11.9% had received at least one dose of the vaccine (95% CI: 10.7–13.1), while 88.1 percent had never been vaccinated (95% CI: 86.8–89.3). The average age of all women at the first vaccination dose was 34.9 years (range: 20–62). The vaccination status of all cases, cervical cancer and H-SIL, correlated by age at diagnosis and age at vaccination, is detailed in Table 1.

Among women with cervical cancer, 96% (95% CI: 94.35–97.29%) had never received any vaccine during their lives. None of the cases with cervical cancer had been vaccinated before diagnosis. The vaccinated women with cervical cancer had been vaccinated after the diagnosis of the cervical lesion, with an average interval of 1872 days (IQR: 546–2799) between the diagnosis and the first dose, and only 3% (95% CI: 1.95–4.56%) had completed the expected cycle. The mean age at diagnosis of vaccinated women was 35.3 years (range: 22–61) while the average age at vaccination was 39.5 years (range: 25–68).

Among women diagnosed with H-SILs, 85% (95% CI: 83.32–86.58%) of them had never received a vaccination against HPV. The vast majority of patients had been vaccinated after the diagnosis of cervical lesions; the average number of days between diagnosis and first vaccination dose was 1294 (IQR: 356–1814) with a median of 623 days. The mean age at diagnosis of vaccinated women was 34.9 years (range: 20–62), while the average age at vaccination was 37.1 years (range: 14–69).

We observed 10 H-SIL cases in women who were previously vaccinated (0.53%, 95% CI: 0.25–0.97%). A total of 275 women started the vaccination course after diagnosis of H-SIL (14.5%, 95% CI: 12.91–16.13%), and only 113 (41%, 95% CI: 35.22–47.16%) of these women started the course within 1 year of diagnosis (Table 1). In six cases, the time between vaccination and diagnosis was on average 7 years; in the other four cases, less than 15 months had elapsed between the completion of the vaccination cycle and the diagnosis of the disease. The woman’s average age at vaccination was 19.5 ± 4.1 years, and the average age at diagnosis was 28 ± 4.1 years. 

There were statistically significant differences in the frequency of high-grade squamous intraepithelial lesions (H-SIL) and cervical carcinoma between vaccinated and unvaccinated women.

Among vaccinated women, 10 H-SIL cases (3.2%) were observed, compared to 1891 cases (80.9%) among unvaccinated women. The relative risk (RR) of developing H-SILs in vaccinated women compared with unvaccinated women was 0.013 (95% CI 0.007–0.024), indicating a 98.7% reduction in risk associated with vaccination. The Pearson chi-square test showed a highly significant difference between the two groups (χ^2^(1) = 826.46; *p* < 0.001), confirmed by the two-sided Fisher’s exact test (*p* < 0.001).

No carcinoma cases were observed among vaccinated women, whereas 751 cases (32.1%) occurred among unvaccinated women. The RR, calculated using the Haldane–Anscombe correction, was 0.008 (95% CI 0.001–0.057), corresponding to a 99.2% reduction in risk among vaccinated women compared to unvaccinated women. The Pearson chi-square test indicated a highly significant difference between groups (χ^2^(1) = 139.28; *p* < 0.001), confirmed by the two-sided Fisher’s exact test (*p* < 0.001).

## 4. Discussion

Our data confirm that the HPV vaccine is an effective but underutilized prevention tool to reduce the incidence of cervical and other HPV-related cancers [24,25]. We found statistically significant differences in the frequency of high-grade squamous intraepithelial lesions and cervical carcinoma between vaccinated and unvaccinated women. Very few women were vaccinated at the recommended age. Most vaccinations occurred after diagnosis, particularly among older women. The problems related to suboptimal use of the HPV vaccine were analyzed separately for primary, secondary, and tertiary prevention of HPV-related disease.

### 4.1. Primary Prevention/HPV Vaccines

In Italy, vaccines against HPV are offered free of charge to children aged 11 to 14. In our province, the HPV vaccine is free for birth cohorts from 1996 for females and from 2003 for males, who are not yet vaccinated. It is also offered on co-payment for other cohorts of females and males and in subjects considered at risk for pre-neoplastic lesions.

The model [26] and observational data [27] indicate that HPV vaccination can have a significant impact on HPV-related outcomes at a population level, with a reduction in cervical cancer and incidence of H-SIL in young women. In our study, we observed that, since the vaccine became available in 2008, none of the women who were vaccinated against HPV developed cancer until 2020, while 1575 high-grade lesions were recorded among unvaccinated women.

The cases of H-SILs recorded in women already vaccinated were detected in subjects that, at the time of vaccination, were already beyond the age recommended by the guidelines. All of them received the 4-valent vaccine. Four cases developed lesions less than 15 months after administration of the vaccine, and we do not consider this a vaccine failure because the infection was in all probability already present but undiagnosed. In the other six cases who developed an H-SIL, the infection was presumably caused by virus subtypes not included in the vaccine.

Vaccination coverage remains particularly low in our province [28], with 26.7% of the female population born between 1996 and 2012 having received a full immunization and 54.6% at least one dose, while 24.4% of the males born between 2003 and 2012 are fully immunized and 40.5% received at least one dose of vaccine.

The Provincial Health Authority (ASP) of Catania began a significant prevention campaign in 2017, offering HPV vaccination to both male and female adolescents. A gradual increase in vaccinated women has been observed since 2018 (Figure 2), and a peak was recorded during the period of the COVID-19 pandemic, where a decrease in screening is likely to be matched by an increase in vaccinations. We do not know whether this was due to a greater sensitivity towards vaccinations or because it was considered the only preventive method available at that time. Women with cervical abnormalities responded more easily to vaccine supplies and peaked in numbers before the general population.

Adherence to vaccination has been improved by the publication of national guidelines in 2020. The vaccination promotion campaign was subsequently accompanied by a catch-up vaccination program which will continue in the coming years. It is necessary to increase vaccination coverage in males and to implement cost-effective catch-up vaccination programs for young men [29].

### 4.2. Secondary Prevention/HPV Screening

The long preclinical phase and the possibility of diagnosing and removing precancerous lesions are the strengths of secondary prevention programs. The increase in vaccination coverage and the fraction of women being screened cover different segments of the population but are interrelated.

Italy initiated cervical cancer screening programs in 1989. Screening is performed every 3 years for people aged 24–64 years but the adherence is still insufficient for effective prevention at the population level [30]. In Catania, 50% of women aged 25–64 years have been screened as part of an organized program, while 27% have done so as individual preventive measures; 17% of women have never had a cervical screening, and 14% have had it performed for more than 3 years.

To reduce mortality, more efforts are required to screen all of the population at risk and those who have not been vaccinated. Similarly, women who do not want or cannot undergo screening should be vaccinated.

The screening protocol needs to be modified due to the widespread use of HPV testing for screening, which allows us to lengthen the intervals of the procedure, and due to the introduction of the HPV immunization program.

Countries that have achieved optimal vaccination coverage (e.g., Denmark, Sweden, or Australia) could revisit their HPV screening programs by extending the interval period and reducing costs [31,32,33]. HPV-vaccinated women can expect a significant decrease in their lifetime risk of H-SIL, which would lead to less extensive screening [34,35].

Further research is needed to determine intervals for re-screening HPV-negative women [36,37].

HPV vaccination could, in fact, modify some characteristics of the virus and generate false negatives or false positives in the screening of vaccinated cohorts [38,39]. Correct secondary prevention can only be carried out with validated kits for safe diagnoses, an essential requirement for screening dispensed by the National Health System; otherwise, the only safe possibility for prevention remains the vaccine.

The extension of HPV screening to men, in whom it is currently used for infertility diagnosis, would certainly help in the secondary prevention of HPV-related lesions and cancers.

### 4.3. Tertiary Prevention/Adjuvant HPV Vaccination

According to our study, the adjuvant vaccine is still not in use: only 15% of women with H-SILs were given the HPV vaccine after the diagnosis of the lesion. An average of 592 days elapsed between diagnosis and the first administration of the vaccine.

The use of vaccination in the treatment of H-SILs may reduce the circulation of the virus, contributing to a long-term reduction in the incidence of cervical cancer [40]. When precancerous lesions are found, they are treated with excisional procedures. Full post-conization vaccination, with the 9vHPV, contributed to an additional reduction in the risk of recurrence and suggests a high adjuvant effect of the 9vHPV vaccine [15,16,41]. The precise timing of HPV vaccination to reduce lesion recurrence remains unresolved [42,43].

Finally, it must be remembered that vaccination for people infected with HPV could also reduce the risk of developing a second HPV-related primary cancer [44,45,46,47].

### 4.4. HPV Vaccine Communication Strategies

Primary vaccination is still carried out at a low rate in our province, as are the screening and vaccination of women with H-SILs. This shows that communication is not effective, as it is hard to find another reason for the low use of a safe, effective, easy to administer, and free vaccine that prevents cancers (not only cervix) and genital warts. Only joint action by all healthcare providers can increase immunization coverage. Epidemiologists and public health specialists should guide policymakers to take public actions that favor an increase in vaccination practice; gynecologists, pediatricians, general practitioners, screening workers, and educators need to support individuals to make the correct choices.

Since it is known that the vaccination is more effective in adolescents, it is important to consider appropriate communication channels. Young people have access to a large amount of information through the Internet and social media. Information does not come from parents, teachers, or medical doctors but is absorbed from the environment and, less often, from TV programs.

Considering that teenagers often do not like to talk about their health problems and sexual relationships, it could be useful to launch particular vaccination invitation campaigns by selecting an active influencer on social media as the spokesperson. Numerous studies have been carried out on the ability of social media to influence therapeutic choices [48,49,50,51] but also on the risks arising from incorrect information: using social media to promote the correct choice would be a means to quickly reach a large number of users at a low cost. Vaccination banners posted in places most frequented by adolescents (schools, pubs, discos, gyms, bars) or in shops (clothing, telephone, electric cars for teenagers, motorbikes/scooters) can also be useful.

Lack of information from attending physicians, low confidence in the vaccine among gynecologists, hesitation of women, and ignorance among adolescents and males are challenges that must be overcome [52]. The possible secondary damage due to conization in women, including preterm delivery and low birth weight [53,54] and infertility in men [55], must also be made widely known.

The practice of correctly vaccinating women with H-SILs and the possibility of tertiary prevention, as a support for the treatment of cervical carcinoma, are not yet widespread. Since modern guidelines have added prophylactic vaccination as a post-intervention use of the vaccine, the scientific community should strive to publish and disseminate shared guidelines that define the most effective timing for the administration of the vaccine in subjects who have already had a cancer lesion. All general practitioners and gynecologists should be carefully informed of this evidence.

A vaccine literacy organization, for creating an environment that allows individuals to find and use information and services on vaccines correctly, could help policymakers and individuals make informed decisions about vaccines [56].

In contrast to cervical cancers, the global burden of cancers largely attributable to HPV (vagina, vulva, anus, and some head and neck cancers) is increasing. Since screening with HPV tests is not effective for these tumors as for the cervix, vaccination in both sexes is an irreplaceable weapon for the possible reduction in the incidence of these tumors and their socio-economic burden [57,58].

### 4.5. Limitations

Our study is a monocentric retrospective study that considered the effects of different vaccinations performed at different times, with different characteristics and numbers of doses. We did not record behavioral data from our patients. Future research must prospectively follow-up correctly vaccinated women (by age and dose) to confirm the effectiveness of the nonavalent vaccine against HPV-related lesions, as well as the rarer cervical carcinomas and HPV-related tumors.

## 5. Conclusions

The number of new cases of cervical cancer has shown a reduction in the last 30 years due to effective prevention (first screening and then vaccination), while the death rate has been less reduced than in other cancers [59,60]. In particular, the 9vHPV vaccine has been shown to be effective even in the long term [61].

The study showed that, unfortunately, among women living in the province of Catania, the limited spread of vaccination caused cases of precancerous lesions or tumors, most of which could probably have been avoided. Few cases performed vaccination as secondary prevention after the appearance of cervical lesions, and even less widespread is post-conization vaccination as tertiary prevention.

The number of vaccinated women has increased in the last six years, but we are still far from the target set by the WHO. Where prevention has not been carried out, a decline in incidence and mortality rates has not been observed [62].

Based on our experience we suggest vaccinating women in the hospital when the excision is carried out and encouraging catch-up programs for H-SIL patients to reduce the circulation of the virus and the risk of developing secondary HPV lesions. Improving information aimed at boys; asking for a vaccination certificate when registering at high school or university; and engaging influencers on social networks for vaccination campaigns could help increase vaccination coverage.

## Figures and Tables

**Figure 1 ijerph-22-01844-f001:**
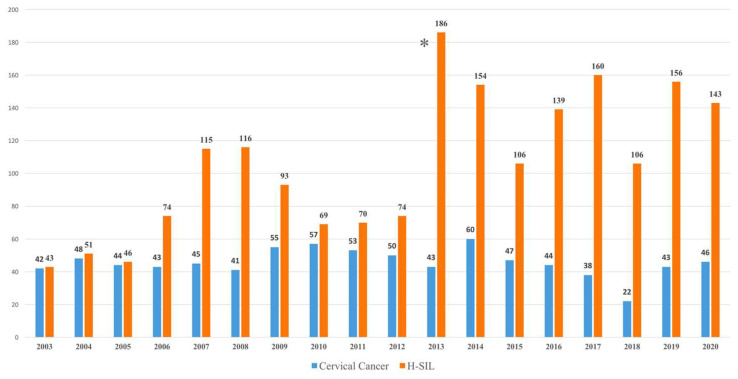
Cervical cancers and High-grade Squamous Intraepithelial Lesions (H-SIL) cases diagnosed in the Province of Catania in the period 2003–2020. * Introduction of cervical cancer screening with HPV tests in addition to the PAP test.

**Figure 2 ijerph-22-01844-f002:**
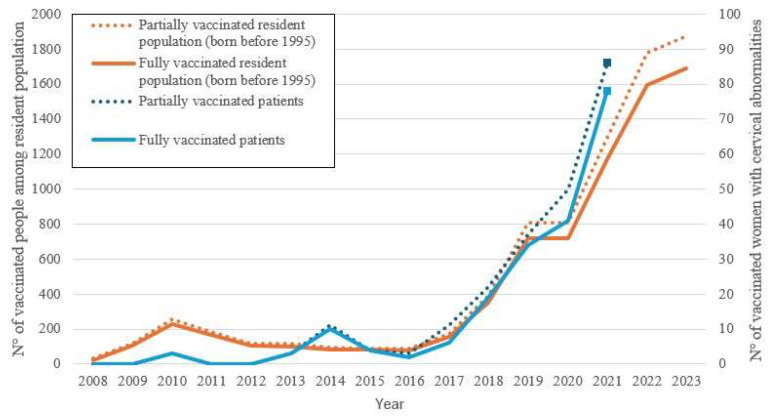
The number of fully and partially vaccinated women among the resident population of the Province of Catania born before 1995, and the number of fully and partially vaccinated women with cervical abnormalities (cervical cancer, H-SIL), shown by year of vaccination. The figure shows data from 2008, the year of free introduction for 12-year-olds in our region, as of 31 December 2023 for residents in Catania and as of 31 December 2021 for patients with cervical abnormalities.

**Table 1 ijerph-22-01844-t001:** Vaccination status of cases diagnosed in the Province of Catania in the period 2003–2020 by age at diagnosis, age at vaccination (all cases, cervical cancers, cervical lesions (H-SIL).

	Number	%	Age at Diagnosis			Age at Vaccination		
			Mean	Standard deviation	Median	Mean	Standard deviation	Median
**All cases**								
Population	2652		43.4	13.5	41.3			
Not vaccinated	2337	88.1	44.6	13.6	42.5	/	/	/
Vaccinated	315	11.9	34.9	8.9	32.7	37.3	9.9	36
*Partially* *vaccinated*	*53*	*2*	*37.* *1*	*10.* *1*	*35.* *9*	*41.* *5*	*11.* *4*	*41*
*Fully vaccinated*	*262*	*9.* *9*	*34.* *5*	*8.* *6*	*31.* *9*	*36.* *4*	*9.* *4*	*35.* *5*
**Pre-diagnosis** vaccine *	10	3.5	29.3	6.2	26.8	23.4	6.2	26.8
**Post diagnosis** vaccine	305	96.5	35.1	8.8	32.8	37.7	9.6	37

**Cervical cancers**								
Population	751	28.3	52.6	14.7	51.3			
Not vaccinated	721	96	53.3	14.4	51.8	/	/	/
Vaccinated	30	4	35.3	9.7	32.9	39.5	11.1	37
**Pre-diagnosis** vaccine *	0							
**Post diagnosis** vaccine	30	4	35.3	9.7	32.9	39.5	11.1	37
*Partially vaccinated*	*7*	*0.* *9*	*37.* *7*	*14.* *6*	*33.* *6*	*44.* *3*	*16.* *4*	*43*
*Fully vaccinated*	*23*	*3.* *1*	*34.* *6*	*8*	*32.* *8*	*38.* *1*	*8.* *9*	*37*

**Cervical lesions (H-SIL)**								
Population	1901	71.7	39.8	11.1	38.7			
Not vaccinated	1616	85	40.7	11.2	39.6	/	/	/
Vaccinated	285	15	34.9	8.8	32.5	37.1	9.8	36
**Pre-diagnosis** vaccine *	10	3.5	29.3	6.2	26.8	23.4	6.2	26.8
**Post diagnosis** vaccine	275	96.5	34.9	8.8	32.5	37.1	9.8	36
*Partially vaccinated*	*46*	*16.* *7*	*37*	*9.* *4*	*36.* *5*	*41.* *1*	*10.* *6*	*41*
*Fully vaccinated*	*229*	*83.* *3*	*34.* *8*	*8.* *6*	*32*	*36.* *9*	*9*	*36*
*Complete vaccination, started within one year* *after diagnosis*	*113*	*49*	*34.* *9*	*8.* *2*	*31.* *8*	*34.* *8*	*8.* *2*	*32.* *0*

* A patient received the first dose of the vaccine 2 months before the diagnosis of cervical cancer, probably after performing a screening test and completed the other two doses after diagnosis.

## Data Availability

We are not permitted to share individual data. Aggregated-level data, in the form of counts, rates can only be shared upon express permission from the participating authorities. These data should be requested by contacting the corresponding author.

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
