# Peer review of "HPV Vaccine: An Effective but Underutilized Prevention Tool"

_ijerph, 2025, doi:10.3390/ijerph22121844_

Round 1

Reviewer 1 Report

Comments and Suggestions for Authors

Well written manuscript.

A few quaint sentences need to be rewritten, preferably by a native English speaking person.

For example: Cervical cancer was diagnosed in women who were not vaccinated before the diagnosis. This sentence in the abstract does not make sense.

It would be prudent to split the topic into invasive cervical carcinoma and H-SIL, so that readers do not need to refer back to the beginning of the sentence again and again.  

A paragraph on remedial measures to increase HPV uptake may be added, for the preventive health program in Italy

Author Response

Responses to your valuable comments are indicated in the attached file.

Reviewer 2 Report

Comments and Suggestions for Authors

Journal: IJERPH (ISSN 1660-4601)

Manuscript ID: ijerph-3903575

Type: Article

Title: HPV vaccine: an effective but under-utilized prevention tool

Authors: Vittorio Grieco , Debora Fiorito , Gabriele Giorgianni , Eleonora Irato , Alessia Anna Di Prima , Chiara Chillari , Liliana Mereu , Rosalia Maria Ragusa *

A brief summary

The topic of this article is both timely and highly interesting. The authors address a significant issue, especially considering that the prevention of HPV-related diseases remains undervalued compared to the treatment of these diseases and their associated complications. Nonetheless, extensive amendments and major revisions are necessary to improve the overall quality of the manuscript. My comments are outlined below:

Line 17-18: The sentence is awkward and unclear; it should be reformulated and rewritten.

Throughout the introduction, references are missing in several places, for example:

Line 37-41, Line 42- 46, Line 46-48: References are lacking.

Although the introduction is relatively well-written, it should place greater emphasis on what distinguishes this research from previous studies and highlight its significance. Additionally, several important data points are missing, such as: the HPV vaccination policy in Italy (specifically, for which age groups is the vaccine approved, regardless of whether vaccination is free or paid, which vaccine is currently in use—I assume nonavalent, which is the national HPV vaccination coverage rate, and how Italy compares in this regard with the European average as well as neighboring countries such as Switzerland, Croatia, Slovenia…). All statements and claims must be supported with appropriate references to substantiate them.

Line 51: Which other EU countries are involved (list them); a reference is also missing here.

Line 88-89: This sentence appears odd; is it a repetition of the sentence in lines 81–82?

Line 96: Please clarify the exact number of recommended doses and the corresponding age groups (for example, in our country, two doses are recommended for individuals up to 15 years old, and three doses for those above 15 years).

The same issue persists throughout the Results section: figures (1 and 2) and table 1 are not inserted immediately after they are mentioned in the text. This must be corrected.

Line 124: Please add percentages in addition to the number of cases.

Line 138: The average age at diagnosis is missing, as it was reported in the previous paragraph (Lines 133–134); please include this information.

Line 139: Why are only 10 cases reported? Given that 85% of women (N = 1901) are unvaccinated, as stated in line 135, and therefore 15% vaccinated, the number of vaccinated cases should be higher than 10.

The entire Discussion section needs to be rewritten: unnecessary parts should be removed, the content shortened, and the text presented in a more logical manner. Some specific comments related to the Discussion are as follows:

Line 165-166: On what basis is the claim in this sentence made? If it refers to your own results, it is not appropriate to cite two external references at the end of the sentence.

Line 189: Please specify which countries are being referred to and add the appropriate reference(s).

Line 189-190: Please clarify what is meant by “suboptimal” vaccination coverage; specify the exact percentage currently reported and the percentage required for coverage to be considered optimal.

Line 239: Please specify which countries are being referred to and add the appropriate reference(s).

Line 275: The phrase "...action must be taken..." is noted, and I agree; however, it is not entirely clear what the authors intended to convey with this sentence. Specifically, who is expected to take this action? The government? Public health specialists? All healthcare providers? Please clarify. Moreover, who are the primary HPV vaccinators in Italy: gynecologists, school doctors, pediatricians, epidemiologists, family physicians, or others?

Line 312: It is recommended that the authors explicitly address the limitations and shortcomings of the study and provide recommendations for future research.

The Conclusion should be written with higher quality and greater specificity to reflect the findings of this study, rather than repeating general well-known facts such as the need to vaccinate both sexes.

Lines 348-474: The formatting of the references is inconsistent. All references should be carefully reviewed to ensure strict adherence to the journal’s author guidelines. Additionally, the necessity of references older than 10 years (specifically 2, 3, 14, 27, 36, 37, 43, and 44) should be critically evaluated, and excluded unless they are essential.

Comments on the Quality of English Language

The quality of English throughout the manuscript is suboptimal, with certain sections being inadequate and at times nearly incomprehensible. The manuscript would substantially benefit from further language refinement. It is strongly advised that the authors consider engaging a professional English-language editing service.

Author Response

Thanks for the extensive comments. Detailed replies are listed on a separate file.

Reviewer 3 Report

Comments and Suggestions for Authors

The manuscript presents a retrospective observational study assessing HPV vaccination status among women diagnosed with cervical cancer or high-grade squamous intraepithelial lesions (H-SIL) in Catania, Italy. The topic is highly relevant and aligns with global public health priorities for HPV vaccination and cervical cancer prevention. The study addresses an important gap in understanding HPV vaccine uptake in a specific Italian region and its implications for public health policy. However, several methodological and interpretative weaknesses should be addressed to strengthen the manuscript’s scientific rigour and impact.

  1. The manuscript requires extensive English language editing throughout. Numerous grammatical errors and awkward phrasings impede readability and detract from the professionalism expected in scholarly writing.
  2. Minor grammatical and typographical errors should be corrected throughout (e.g., “underutilized” vs. “under-utilized,” “percent” vs. “%” consistency).
  3. Specify whether ethical approval covered retrospective registry linkage (this is mentioned but could be clarified in the Ethics section).
  4. Include the limitations section explicitly—currently, limitations are mentioned implicitly but deserve clearer acknowledgement (e.g., lack of behavioural data, limited generalizability, absence of control for confounders).
  5. Abstract: Briefly include numerical results (per cent vaccinated, mean age, etc.) for better readability.
  6. Lack of Correlation Between Vaccination Status and Cervical Cancer Outcomes: The study fails to establish a clear and meaningful correlation between HPV vaccination status and the occurrence of cervical cancer or cervical lesions, which should constitute the central focus and conclusion of the research. This significantly limits the study's contribution and relevance to the existing body of literature.
  7. Deficiencies in the Introduction and Methodology: The introduction lacks a comprehensive and critical literature review, and the methodology section is underdeveloped, lacking sufficient detail and clarity regarding study design, data sources, and analytical methods. These omissions hinder reproducibility and the overall interpretability of the work.
  8. Statistical Analysis: The statistical methods are only briefly described, and no inferential tests or confidence intervals are presented. The analysis remains largely descriptive.
  9. Results Presentation: Results are detailed and supported by tables and figures; however, some figures (especially Figure 2) are not clearly labelled or discussed.
  10. Improper Use of AI in the Discussion Section: It is evident that AI-generated content was used extensively in the discussion section without sufficient editing or integration into the authors' own analysis. This results in a superficial and generic discussion that lacks critical engagement with the data and literature. The discussion occasionally reads like a policy commentary rather than a crucial scientific interpretation.
  11. Conclusions: The conclusions are generally appropriate but could be more concise and focused on the study’s empirical findings rather than broad policy suggestions.
  12. Inconsistent and Incomplete Referencing: The references are inconsistently formatted, with several citations lacking necessary details. Moreover, the reference list does not adequately support the claims made in the manuscript, further weakening the scientific foundation of the study.

Author Response

Responses to your valuable comments are indicated in the attached file

Round 2

Reviewer 2 Report

Comments and Suggestions for Authors

Dear authors,

Thank you for revised version of your manuscript, which is now fine.

Author Response

Dear reviewer,                                                                                                           

Thank you for taking your valuable time to review our work

Reviewer 3 Report

Comments and Suggestions for Authors

This manuscript investigates HPV vaccination uptake and its association with the occurrence of cervical cancer and high-grade squamous intraepithelial lesions (H-SILs) among women diagnosed between 2003 and 2020 in the Province of Catania, Italy. The topic is highly relevant, timely, and aligned with global efforts to eliminate cervical cancer. The dataset is valuable, and the findings reinforce the critical public health impact of HPV vaccination.

The manuscript is generally well-written, and the authors have addressed the comments. However, it includes sections that need tightening, clearer reporting, and more cautious interpretations.

Comments

  1. Textual and Terminology Corrections
  • Please replace “under-utilized” with “underutilised” in the title.
  • Please replace “Human Papilloma Virus” with “Human Papillomavirus” throughout the manuscript.
  • Replace “percentage” with “%” for consistency.
  • Replace “arose” with “were detected.”
  • Improve grammar in several sentences (e.g., change “remain very low” to “remained very low”).
  • Remove extra spaces and unnecessary line breaks.
  1. Numerical and Formatting Corrections
  • Correct decimal formatting by replacing “3,5%” with “3.5%”, and apply this formatting consistently across the manuscript.
  • Correct the percentage corresponding to 1901 cases to 71.7% (instead of 71,7%).
  1. Methods & Results Clarifications
  • Clarify how “vaccinated before diagnosis” was verified (e.g., through date comparison between registry entries).
  • Indicate whether HPV genotyping data were available for the breakthrough HSIL cases (women vaccinated before diagnosis).
  1. Statistical Reporting Improvements
  • Provide confidence intervals for proportions and descriptive statistics where appropriate to strengthen statistical transparency.
  1. Study design clarity and limitations

The study is retrospective and observational, relying on record linkage between cancer registries and vaccination databases. However, the manuscript needs a clearer explanation of:

  • Whether there were missing vaccination records (e.g., private clinics, vaccines administered outside the province).
  • Possible misclassification of vaccination status.

This is essential because the conclusion that “None of the vaccinated women developed cervical cancer” could partially reflect incomplete vaccination record capture.

  1. Interpretational caution regarding causality

Some statements imply a strong causal relationship based on very small numbers (e.g., “None of the women vaccinated developed cancer”). Given the retrospective design and the small, vaccinated subgroup, the manuscript must clearly state that:

  • Most vaccinations occurred after diagnosis, particularly among older women.
  • Very few women were vaccinated at the recommended age, limiting the ability to assess true prophylactic effectiveness.
  • The protective effect observed may partly reflect the characteristics of the vaccinated population.
  1. Table 1 requires complete reformatting.

The current Table 1 is difficult to interpret due to misaligned columns, formatting errors, and unclear structure. Some values (e.g., percentages, medians) appear inconsistent or incomplete. A complete revision of Table 1 is essential.

  • Reformat the table into clear sections (All cases / Cervical cancer / H-SIL).
  • Standardise decimal notation (use periods, not commas).
  • Ensure totals and percentages match.
  1. Figure interpretation requires improvement.

Figure 2 is described, but the manuscript lacks:

  • A clear explanation of how population vaccination coverage was calculated.
  • Clarification that the vaccinated population includes multiple birth cohorts.
  • Additionally, the figure caption should include units, definitions, and time frames.
  1. Discussion: overly long and somewhat narrative

The discussion includes extended commentary on communication strategies, social media, screening organisations, and influencers. While relevant, these sections should be tightened and focused more directly on the study's findings and their implications. A more concise and evidence-based discussion would improve the manuscript’s scientific rigour.

10. Conclusions:

The conclusions are broadly appropriate; however, they would benefit from being more concise and more directly aligned with the study’s empirical findings, rather than expanding into broader policy recommendations. Please avoid adding any new references in the conclusion section.

Author Response

Dear reviewer,                                                                                                           

Thank you for taking your valuable time to review our work

We appreciated the comments. The manuscript has been extensively reviewed and modified.

Detailed replies are listed below under each item.
